# TOWARDS SELF-SUPERVISED LEARNING OF GLOBAL AND OBJECT-CENTRIC REPRESENTATIONS

**Federico Baldassarre & Hossein Azizpour**
KTH - Royal Institute of Technology, Stockholm, Sweden
{fedbal,azizpour}@kth.se

## ABSTRACT

Self-supervision allows learning meaningful representations of natural images, which usually contain one central object. How well does it transfer to multi-entity scenes? We discuss key aspects of learning structured object-centric representations with self-supervision and validate our insights through several experiments on the CLEVR dataset. Regarding the architecture, we confirm the importance of competition for attention-based object discovery, where each image patch is exclusively attended by one object. For training, we show that contrastive losses equipped with matching can be applied directly in a latent space, avoiding pixel-based reconstruction. However, such an optimization objective is sensitive to false negatives (recurring objects) and false positives (matching errors). Careful consideration is thus required around data augmentation and negative sample selection.

## 1 INTRODUCTION

Self-supervised approaches show great promise for learning meaningful representations of unlabeled data that transfer well to downstream tasks (Chen et al., 2020; Grill et al., 2020; Li et al., 2021; Caron et al., 2021; Chen & He, 2021). While most methods produce dense global features, learning structured representations of objects remains an open challenge (Burgess et al., 2019; Greff et al., 2019; Locatello et al., 2020). Such an explicit representation would greatly benefit tasks that involve sparse causal reasoning and modeling physical dynamics (Ha & Schmidhuber, 2018; Kipf et al., 2020; Hafner et al., 2021). We argue that object-centric representations and global scene representations are complementary and should be learned jointly through self-supervision. Thus, the proposed approach combines an attention-based object discovery pipeline and matching contrastive losses in latent space. In this context, we discuss key aspects around architecture and optimization. Here follows a brief overview of relevant related work which is further expanded in Appendix C.

**Self-supervised learning.** Early techniques involved *pretext tasks*, e.g. colorization, inpainting, or spatial reasoning (Doersch et al., 2015; Zhang et al., 2016; Pathak et al., 2016; Noroozi & Favaro, 2016; Gidaris et al., 2018). Then, the focus shifted to learning in a latent space, e.g. through contrastive losses (van den Oord et al., 2018; He et al., 2020; Chen et al., 2020), negative-free similarity (Grill et al., 2020; Chen & He, 2021), online clustering and self-distillation (Caron et al., 2020; 2021; Li et al., 2021). Recent works formulate patch-based losses (Li et al., 2021; Wang et al., 2021), yet the resulting embeddings remain redundant. How to summarize dense features into a sparse representation is an active area of research (Hinton, 2021). A key components of all these methods is data augmentation. In particular, *multi-crop* can be regarded as an implicit suggestion of *objectness*, as it encourages learning similar representations between the global view of an object and its cropped parts. However, this strategy may struggle if images contain more than one object.

**Object discovery.** A plethora of methods for unsupervised object discovery in images and videos exists (Greff et al., 2019; Burgess et al., 2019; Engelcke et al., 2019; Kipf et al., 2021). The learning objective often involves reconstruction or future prediction, which may struggle with real-world textures (Ha & Schmidhuber, 2018), or latent-space losses (Doersch et al., 2015; Kipf et al., 2020). Racah & Chandar (2020) and Löwe et al. (2020) extract object tokens from video frames for time-contrastive learning. However, the former requires additional regularization to encourage diversity in CNN-based tokens, and the latter merges tokens into a global representation before a frame-wise loss. In contrast, we avoid additional diversity regularizers by using Slot Attention (Locatello et al., 2020) and we apply a contrastive loss directly to object tokens using a matching algorithm.

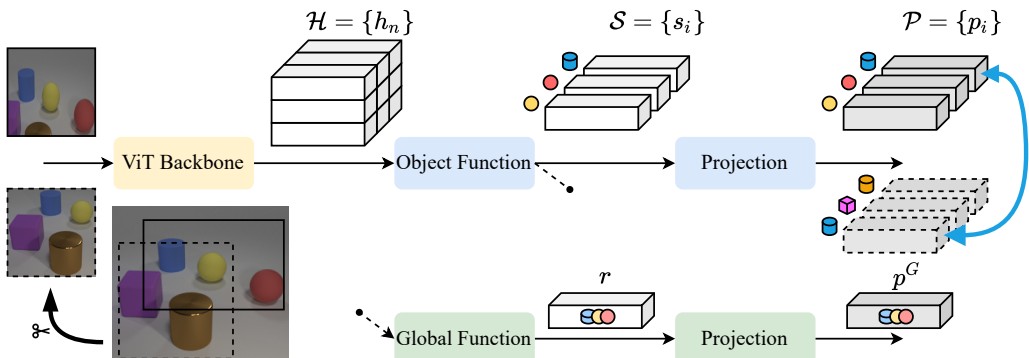

Figure 1: Self-supervised architecture for object discovery. An input crop is processed with a ViT backbone to extract patch features, which are aggregated into object tokens via query-based attention. A matching contrastive loss compares object tokens from different crops. Similarly, a global branch pools object tokens to produce a scene representation, which is contrasted to other images.

## 2 METHOD

Our goal is to learn object-centric representations in a self-supervised fashion, i.e. without explicit annotations of categories or positions. Specifically, given an image we wish to produce a *global representation* that captures the entire scene and its layout, and a set of *object representations* that describe individual objects in isolation, e.g. color and shape but not position. To this end, we discuss key aspects that should be considered when adapting self-supervised learning to object-centric scenarios: inductive biases in the architecture, choice of optimization objective, and data augmentation.

### 2.1 OBJECT-CENTRIC ARCHITECTURE

The typical self-supervised architecture, e.g. (Chen et al., 2020), comprises: a *backbone* that extracts spatial features, a *representation function* that aggregates information into a global feature vector, and a *projection head* that is discarded after training. For object-centric learning, the representation function shall, in addition, output a set of vectors $\mathcal{S} = \{s_i\}$ commonly termed *slots* or *object tokens*.

**Backbone.** We use a vanilla Vision Transformer (ViT) (Vaswani et al., 2017; Dosovitskiy et al., 2020), though any vision backbone, e.g. a CNN, would suffice. First, an input image $x$ of size $H \times W$ is split into $N = HW/P^2$ patches of size $P \times P$. Then, several blocks of multi-head self-attention with positional encoding produce a set $\mathcal{H} = \{h_n\}$ of $D$-dimensional *patch tokens*.

**Object function.** At the core of the architecture, the *object representation function* extracts a set $\mathcal{S} = \{s_i\}$ of *object tokens* from $\mathcal{H}$. For this, several implementations exist, e.g. channel-wise splitting (Kipf et al., 2020; Racah & Chandar, 2020), iterative variational inference (Greff et al., 2019; Burgess et al., 2019), or attention with *query tokens* (Carion et al., 2020; Locatello et al., 2020; Löwe et al., 2020). In this work, we focus on attention mechanisms and remark the importance of *competition* to induce query specialization and promote diversity in the resulting object tokens. Specifically, we consider Slot Attention (Locatello et al., 2020), which operates similarly to cross-attention (Vaswani et al., 2017), but applies $\mathrm{softmax}$ normalization in the query dimension. Formally, the attention weights for a query token $q_i$ are: $\alpha_{in} = \exp(\phi(q_i, h_n)) / \sum_{i'} \exp(\phi(q_{i'}, h_n))$; where $\phi$ is a compatibility function. This way, each query competes for attending to a specific patch, which reflects an "exclusivity" inductive bias in images and facilitates learning object-centric representations. Furthermore, Slot Attention is applied iteratively with GRU (Cho et al., 2014) updates, which allows object tokens to specialize conditioned on the input. In contrast, vanilla cross attention defines $\alpha_{in} = \exp(\phi(q_i, h_n)) / \sum_{n'} \exp(\phi(q_i, h_{n'}))$ and lacks a competition-inducing mechanism. As a result, cross-attention tokens may fail to selectively represent a single object as shown in Fig. 3.

**Projection heads.** To accommodate the self-supervised losses described below, a simple MLP projects $s_i$ to $p_i$, and a global branch is attached to the object representation function (Figure 1). The global branch operates on a permutation-invariant element-wise aggregation of object tokens, i.e. $\mathrm{avg}(\mathcal{S})$, to produce the global representation $r$, and is followed by a MLP head that outputs $p^G$.

## 2.2 Optimization objective.

We train these models in an unsupervised fashion with a combination of *global* and *object-wise losses* that operate on the latent projections $\boldsymbol{p}^G$ and $\{\boldsymbol{p}_i\}$ respectively. In contrast to autoencoder-based methods (Kingma & Welling, 2014), we directly optimize for the desired objective and avoid introducing a decoding stage and pixel-wise comparisons (Ha & Schmidhuber, 2018; Locatello et al., 2020; Hafner et al., 2021; Kipf et al., 2021). While the adaptation of contrastive losses to global image representations is straightforward, object-wise losses requires careful consideration.

**Global loss.** Let us first review the contrastive loss formulation popularized in SimCLR (Chen et al., 2020). We consider a batch $\{\boldsymbol{x}_{ab}\}$ containing two augmentations $a \in \{0, 1\}$ of each image $b = 1, \ldots, B$. Based on its projection $\boldsymbol{p}^G_{ab}$, each input $(a, b)$ should be classified as its other augmented version $(\bar{a}, b)$, i.e. the *positive sample*, against the remaining $2B - 2$ inputs, i.e. the *negative samples*:

$$\mathcal{L}^{\mathrm{G}}(a, b) = -\log \frac{\exp\left(\cos\left(\boldsymbol{p}^G_{ab}, \boldsymbol{p}^G_{\bar{a}b}\right)/\tau\right)}{\sum_{(a', b') \neq (a, b)} \exp\left(\cos\left(\boldsymbol{p}^G_{ab}, \boldsymbol{p}^G_{a'b'}\right)/\tau\right)}, \tag{1}$$

where $\tau \in \mathbb{R}^+$ is a hyperparameter. This formulation encourages the global representation to capture an image as a whole, including e.g. its layout, which sets it apart from other images in the batch.

**Object loss.** In a similar spirit, our object loss describes a pseudo classification task where each object token of $\boldsymbol{x}_{ab}$ should identify a positive sample from object tokens of $\boldsymbol{x}_{\bar{a}b}$ against a set of unrelated negative ones. Differently from $\mathcal{L}^{\mathrm{G}}$, the correspondence between object tokens is not predetermined. Thus, for each $(a, b)$, we apply the Hungarian algorithm (Kuhn, 1955) to obtain the bipartite matching $\sigma : i \to j$ between the tokens $\{\boldsymbol{s}_{abi}\}$ and $\{\boldsymbol{s}_{\bar{a}bj}\}$ that minimizes $\sum_i \cos(\boldsymbol{s}_{abi}, \boldsymbol{s}_{\bar{a}b\sigma(i)})$. By indicating with $\mathcal{N}(a, b, i)$ the set of negative samples, the loss for a single token can be written as:

$$\mathcal{L}^{\mathrm{CTR}}(a, b, i) = -\log \frac{\exp\left(\cos\left(\boldsymbol{p}_{abi}, \boldsymbol{p}_{\bar{a}b\sigma(i)}\right)/\tau\right)}{\sum_{(a', b', i') \in \{(\bar{a}, b, \sigma(i))\} \cup \mathcal{N}(a, b, i)} \exp\left(\cos\left(\boldsymbol{p}_{abi}, \boldsymbol{p}_{a'b'i'}\right)/\tau\right)}. \tag{2}$$

Crucially, this contrastive formulation depends strongly on how negative samples are chosen. Here, we describe two strategies for populating $\mathcal{N}$, namely CTRALL and CTRIMG, and a negative-free alternative termed COSSIM. The most straightforward strategy, designated as CTRALL, considers all objects from all images in the batch as negatives, i.e. $\mathcal{N}^{\mathrm{ALL}}(a, b, i) = \{(a', b', i')|(a', b', i') \neq (a, b, i) \wedge (a', b', i') \neq (\bar{a}, b, \sigma(i))\}$. However, in many real-world datasets, the same object appearing in unrelated images would produce several *false negatives*, thus forcing the model to remember contextual information to differentiate between instances (Figure 2). To drastically reduce of number of false negatives, CTRIMG isolates each pair of augmented images, i.e. $\mathcal{N}^{\mathrm{IMG}}(a, b, i) = \{(a', b, i')|(a', i') \neq (a, i) \wedge (a', i') \neq (\bar{a}, \sigma(i))\}$. Yet, duplicate instances within the same image would still be contrasted as negatives (Figure 2). COSSIM is a negative-free alternative inspired by BYOL and SimSiam that only maximizes the similarity of matching tokens:

$$\mathcal{L}^{\mathrm{SIM}}(a, b, i) = -\cos\left(\boldsymbol{p}_{abi},\ \mathrm{stopgrad}\left(\boldsymbol{s}_{\bar{a}b\sigma(i)}\right)\right). \tag{3}$$

To prevent collapse to trivial solutions, BYOL prescribes a teacher-student setup with momentum updates (Grill et al., 2020), while SimSiam trains a single model in asymmetric siamese configuration (Chen & He, 2021). In our experiments, we observe that the global loss $\mathcal{L}^{\mathrm{G}}$ in combination with the $\mathrm{stopgrad}$ operator in $\mathcal{L}^{\mathrm{SIM}}$ are sufficient for learning object-centric representations.

## 2.3 Augmentations for multi-object datasets

Another issue associated to object losses is the occurrence of *false positives* in the matching step if the augmentation strategy includes heavy cropping. For natural images containing one prominent object, multi-crop manages to capture part-whole relationships (Chen et al., 2020; Chen & He, 2021; Caron et al., 2021). However, tight crops of multi-object scenes as in Fig. 2 may include unrelated instances that are considered a positive match nonetheless, hindering the learned representation (Hinton, 2021). For this reason, a trade-off should be drawn between the global contrastive loss, which calls for more extreme crops, and the object loss, which requires more stability to avoid matching mistakes. In all experiments, we avoid extreme crops by setting a minimum and maximum threshold so that each scene is almost entirely visible. Other augmentation techniques may be used freely, e.g. rotations, distortions, and color jittering. As such, both representation goals are achieved: global features will capture objects and their layout to be able to distinguish different scenes, and object features will naturally cluster objects that appear similar up to some given perturbations.

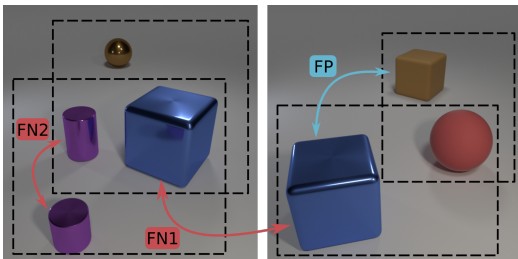 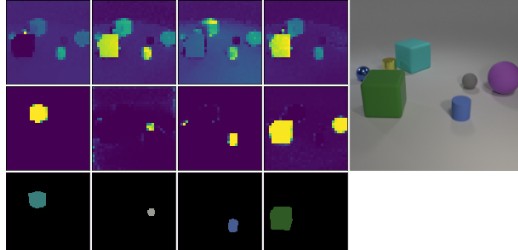

Figure 2: Contrastive losses push **negatives** apart and **positives** closer. False negatives occur if the same object appears in unrelated images (**FN1**, affects CTRALL) and if different crops of the same image contain duplicate instances (**FN2**, affects both CTR*). False positives are due matching mistakes across crops (**FP**, affects all losses).

Figure 3: Per-object attention maps obtained through Cross Attention (top) and Slot Attention (middle) vs. ground-truth masks (bottom). Both methods clearly focus on objects. Cross attention does not encourage specialization (columns 2 and 4). Slot attention may merge spatially-distant objects in the same token (column 4).

## 3 EXPERIMENTS

Our goal is to empirically evaluate whether the proposed architecture and training objective capture individual objects, and whether the learned object features are more suitable than global features for object-centric tasks. We further analyze the role of $\mathcal{L}^{\mathrm{G}}$ and cropping in a set of ablation studies.

**Setup.** The backbone is a 2-layer ViT with $4 \times 4$ patches and 256-dim embeddings. Slot attention uses 11 learned query tokens and one GRU iteration. Alternatively, 2 layers of cross attention with 4 heads are applied. All experiments use $128 \times 128$ images from CLEVR (Johnson et al., 2017; Kabra et al., 2019) augmented with cropping, horizontal flipping, and color jittering. The Adam optimizer (Kingma & Ba, 2015) minimizes global and object losses on the training set, while hyperparameters are optimized w.r.t. validation losses. Metrics are reported on a held-out test split. Details in App. A.

**Object segmentation.** First, we wish to determine whether object tokens attend to actual objects. We extract binary masks from the attention maps of a trained model, as shown in Figure 3, and measure the Intersection over Union (IoU) w.r.t. a ground-truth segmentation. Details in Appendix A. The results in Figure 4 (left) show that Slot Attention tends to focus on individual objects more than Cross Attention, which instead struggles to segment instances. This validates the importance of a competition-inducing mechanism for the object representation function. Moreover, $\mathcal{L}^{\mathrm{SIM}}$ and $\mathcal{L}^{\mathrm{CTRIMG}}$ produce the highest IoU, which we attribute to the reduced occurrence of false negatives.

**Learned representation.** We then evaluate the object representations by training *linear probes*, i.e. $\mathrm{sigmoid}\,(\max\{\boldsymbol{W}\boldsymbol{s}_i + \boldsymbol{b}\})$, to predict VQA-style questions in the form "is there a *(size, color, material, shape)* object in the image?" (Zhang et al., 2016; Anand et al., 2019). For a SimCLR-equivalent baseline, the global branch is attached directly to the backbone, the loss includes only $\mathcal{L}^{\mathrm{G}}$, and the probe is simply $\mathrm{sigmoid}\,(\boldsymbol{W}\boldsymbol{r} + \boldsymbol{b})$. Figure 4 (right) reports Average Precision (AP) for various configurations (more in Appendix A.6). The main observation is that object tokens are superior to a global representation for object-centric downstream tasks, albeit simple as this one. Furthermore, Slot Attention produces the most informative features if false negatives are reduced.

**Additional studies.** For both unsupervised segmentation and linear VQA evaluation, we also train models *without* the global loss component to investigate its role in the optimization. Without it, we observe a performance drop for all combinations of attention and losses (Fig. 4), which indicates that $\mathcal{L}^{\mathrm{G}}$ is an important regularizer. We remark that global and object representations are complementary and their joint optimization yields superior representations. Here, the interesting behavior of CTRALL deserves additional discussion: on the one hand, attention maps completely fail to segment objects, on the other, VQA performance remains high. Recall from Section 2.2 that this loss includes *all* tokens from *all* images in the batch as negatives. Minimizing this loss requires attending to the entire scene and incorporating contextual information in each object token, which would otherwise be indistinguishable from potential duplicates. This explains why VQA performance does not plummet even though the learned representations are not focused on single instances. We suspect that these models would eventually fail harder tasks e.g. attribute prediction, counting and reasoning.

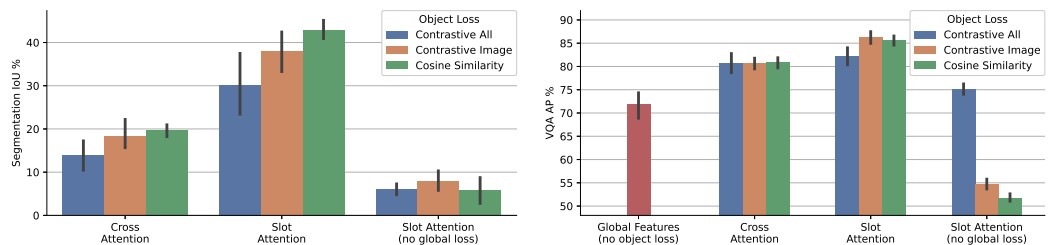

Figure 4: Evaluation of learned features on unsupervised segmentation and linear-probe VQA. For segmentation, object attention maps are binarized and matched with ground-truth masks, excluding the background. For VQA, a linear model predicts binary questions from the learned object tokens. We report Intersection over Union and Average Precision as the mean and std of multiple runs.

Last, we evaluate the effect of crop scale by sweeping over a grid of minimum and maximum relative area chosen in $[0.1, 0.3, 0.5, 0.8, 0.9, 1.0]$. As discussed in Section 2.3, we expect a graceful degradation of the quality of the object features associated to wider crop ranges. However, we observe that a narrower crop range actually yields worse performance for both segmentation and VQA (see tables in Appendix B.3). It remains to be assessed whether this result is specific to CLEVR or a general property of global contrastive losses when combined with object-wise losses.

## 4 CONCLUSION

**Summary.** In this study, we discuss an approach for learning image-level and object-centric representations in a self-supervised fashion. The key components are a competition-based attention mechanism, such as Slot Attention, combined with contrastive losses applied directly in the global and object latent spaces. Compared to previous work, our approach does not rely on pixel reconstruction (Greff et al., 2019; Locatello et al., 2020) which may become challenging on natural images, and does not require regularization terms (Racah & Chandar, 2020) to encourage token diversity thanks to its architecture and joint optimization objectives. Supported by a series of small experiments on the CLEVR dataset, we study the interplay of architecture components, namely, Cross Attention and Slot Attention, and alternative formulations for the object loss. A first takeaway is that competition is necessary for attention-based object discovery, even when training in a contrastive fashion. Alternatives to Slot Attention such as online clustering and dynamic routing may be of interest for future research, especially if they enable discarding unmatched tokens. Also, we observe that joint global/object optimization improves the learned object representations by offloading contextual information to the global representation. The resulting object features show promising results on an object-centric downstream task, which remain to be validated in more advanced scenarios.

**Limitations and future work.** For simplicity, the patch size is maintained constant at $4 \times 4$ pixels, resulting in blocky segmentation maps which are in turn a symptom of a too-coarse resolution. Adopting pyramidal architectures such as Swin, MViT, or FPN (Liu et al., 2021; Fan et al., 2021; Lin et al., 2017) may be necessary to capture greater object diversity and reduce computational cost. Our evaluation protocol, namely, unsupervised segmentation and linear-probe VQA, is designed for simplicity and ease of assessment. More rigorous metrics, e.g. compactness and modularity (Racah & Chandar, 2020), and more extensive datasets, e.g. (Kabra et al., 2019), are a natural follow-up. The standard dot-product attention offers limited control on its attention. In CLEVR, we observe multiple cases of spatially-separate objects being merged based on appearance (Fig. 3). Alternatives to $\mathrm{softmax}$ could improve both locality and sparsity (Niculae & Blondel, 2017; Martins et al., 2021). While this work focuses on images, we are convinced that the temporal dimension contains important clues about objectness and causality. Building upon recent work (Kipf et al., 2021; Hafner et al., 2021), such inductive biases may be integrated in the contrastive loss for learning on video data. Finally, although the current objective is to learn "flat" sets of object representations, our human representations are undoubtedly hierarchical. A clear definition of such hierarchy and quantitative ways of evaluating the learned representations will be critical for advancing computer vision research.

Code, datasets, and notebooks are available at `github.com/baldassarreFe/iclr-osc-22`

ACKNOWLEDGEMENTS

This work was funded by the Swedish Research Council (Vetenskapsrådet) project 2017-04609 and by the Wallenberg AI, Autonomous Systems and Software Program (WASP).

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

# Appendix

## A  ADDITIONAL DETAILS

### A.1  DATASETS

**CLEVR with masks.**   We use the CLEVR with masks dataset provided by (Kabra et al., 2019). It contains 100k images of size $320 \times 240$. We use the first 70k for self-supervised training, and the following 15k for monitoring the losses during training and performing hyperparameter optimization. The last 15k samples are used for the final evaluation of unsupervised segmentation (all 15k) and linear-probe VQA (10k for training the linear model, 5k for reporting average precision).

For self-supervised training, apply the following transformations:

1. Random horizontal flip (left-right)
2. Random crop: a rectangle is cropped at random provided that its aspect ratio and area are within given thresholds
3. Resize to $128 \times 128$
4. Color jittering: brightness, saturation, hue
5. RGB channels normalization with mean $[0.485, 0.456, 0.406]$ and standard deviation $[0.229, 0.224, 0.225]$

### A.2  ARCHITECTURE

#### A.2.1  BACKBONE.

- The backbone is a Vision Transformer with 256-dimensional embeddings, 2 layers and 4 attention heads.
- Positional embeddings are learned (a $32 \times 32$ grid for $4 \times 4$ patches) and applied at every layer.
- Layer normalization is applied before each self attention and MLP layer.
- The MLP layers use GeLU activations and have a 512-dimensional hidden space.
- No CLS token is used in the backbone as output patch features are used as inputs to the attention-based object representation function. the only exception is the global-only configuration, where the object branch is detached and the CLS token represents the input to the global representation function.

#### A.2.2  QUERY TOKENS

The object representation function aggregates patch tokens into object tokens by applying slot or cross attention. Query tokens represent the other input to the attention mechanism. We explore several approaches to obtain query tokens:

- **Learned:** the standard query tokens used throughout the paper. These are simply learned embeddings in fixed number, i.e. 11 for CLEVR.
- **Sampled:** query tokens are sampled from a distribution, which allows adjusting their number at runtime:
  - **Single Gaussian:** all query tokens are sampled from a single Gaussian distributions with learned mean and standard deviation;
  - **Uniform mixture of Gaussians:** we learn the parameters of a fixed number of Gaussian components (mean and std) and uniformly sample query tokens from the mixture distribution.
- **K-means initialization:** we initialize the query tokens by applying K-means clustering with Euclidean norm to the patch tokens of the backbone. K can be determined at runtime.

Across all experiments, only learned query embeddings result in stable training. Single-gaussian sampled queries often degenerate into a constant. The uniform mixture suffers from either too few components (each component is oversampled and the queries do not specialize), or too many components (they fail to train as they are selected too rarely). K-means initialization gives poor results, possibly due to a mismatch between the patch tokens where K-means is applied and actual query-key pairs that are produced with linear projections for the attention.

### A.2.3 OBJECT REPRESENTATION FUNCTION

**Cross attention.** A standard cross attention module with pre-normalization. Activation and MLP dimension are the same as the backbone. We experiment with 1-2 layers 4 attention heads. Each layer has its own weights.

**Slot attention.** Slot attention acts as standard cross attention layer but applies $\mathrm{softmax}$ along the query axis (columns). Attention weights are then normalized along the key axis (rows) by dividing each value by the sum of its row, this ensures that each rows sums to 1 as in regular attention. If multiple iterations are performed, the weights of the key-query-value projections and of the GRU unit are reused. Together with having a single attention heads, this makes slot attention a more lightweight module in terms of learnable parameters.

### A.3 LOSSES

A vanilla implementation of the object loss functions described in the main text would require computing the pairwise cosine similarity of $2 \times \mathrm{batch_size} \times \mathrm{num_objects}$ vectors of size D. On top of this, $2 \times \mathrm{batch_size}$ bipartite matching problems of size $\mathrm{num_objects} \times \mathrm{num_objects}$ would be computed. Instead, we leverage symmetry between augmentations to reduce the computational cost of evaluating the losses. See the provided code for further details. Overall, the cost of evaluating the loss function over batches up to $2 \times 64$ images is minimal compared to a forward pass.

### A.4 TRAINING AND HYPERPARAMETERS

We train all models using the Adam optimizer (Kingma & Ba, 2015) with weight decay. We linearly increase the learning rate from 0 to 0.0007 for the first two epochs, then use cosine decay to gradually reduce the learning rate to 0.0003 in the span of 8 epochs. These hyperparameters were empirically chosen by analyzing the loss curves on the validation split during early experiments and then kept fixed throughout.

### A.5 SEGMENTATION MASKS

Once a model is trained, attention maps are extracted from its first layer of cross attention or its first iteration of slot attention. Attention maps are individually binarized using Otsu thresholding. As a result, it is possible for segmentation masks to overlap, thought this does not provide any benefit for the evaluation.

Computing the Intersection over Union w.r.t. ground-truth masks requires first solving yet another bipartite matching problem, where each ground-truth is assigned to a predicted mask such that the total IoU is maximized. If the image contains fewer object than the segmentation masks obtained from query tokens, the remaining predictions are discarded. When computing the average IoU for an image, the background mask is ignored, and unmatched predictions are ignored. Effectively, IoU is computed from the perspective of the ground-truth.

## A.6 LINEAR PROBES FOR VQA

For the evaluation of learned features, a simple linear model applied object tokens to project the 256-dim embedding into a 96-dim vector. These 11 vectors are then max-pooled into a single image-wise vector that represent the pre-sigmoid predictions to the 96 possible binary VQA questions. All combinations of size, color, material, and shape attributes and their relative counts in the test dataset are summarized in Table 1. The weight of the linear model are trained for 10k steps using Adam with weight decay.

Table 1: Percentage of test images that contain at least one object of the indicated (size, color, material, shape), i.e. class imbalance for the VQA task used for linear evaluation.

| Size | Material | Color / Shape | A | B | C | D | E | F | G | H |
|------|----------|---------------|-----|-----|-----|-----|-----|-----|-----|-----|
| Large | Metal | Cube | 6.5 | 6.3 | 6.7 | 6.7 | 6.1 | 6.7 | 6.3 | 6.6 |
| | | Cylinder | 6.1 | 6.1 | 5.9 | 6.2 | 6.1 | 5.6 | 6.1 | 5.8 |
| | | Sphere | 6.2 | 6.5 | 6.2 | 6.3 | 6.0 | 6.0 | 6.1 | 6.3 |
| | Rubber | Cube | 6.4 | 6.6 | 6.6 | 6.2 | 6.6 | 6.5 | 6.3 | 6.6 |
| | | Cylinder | 6.1 | 5.9 | 6.2 | 6.3 | 6.2 | 5.9 | 6.0 | 5.8 |
| | | Sphere | 6.1 | 6.3 | 5.7 | 6.4 | 6.2 | 6.4 | 6.3 | 6.2 |
| Small | Metal | Cube | 6.9 | 6.6 | 6.8 | 6.3 | 6.6 | 6.6 | 6.6 | 6.5 |
| | | Cylinder | 7.3 | 6.9 | 6.7 | 6.9 | 7.3 | 7.1 | 6.9 | 6.7 |
| | | Sphere | 6.7 | 7.5 | 7.2 | 7.1 | 7.0 | 6.8 | 6.6 | 6.7 |
| | Rubber | Cube | 6.6 | 6.8 | 6.5 | 6.8 | 6.7 | 7.0 | 6.9 | 6.8 |
| | | Cylinder | 7.0 | 7.4 | 7.0 | 6.9 | 6.6 | 6.9 | 7.1 | 7.8 |
| | | Sphere | 6.8 | 6.6 | 6.9 | 6.8 | 6.8 | 7.3 | 6.9 | 6.8 |

# B  ADDITIONAL RESULTS

## B.1  QUANTITATIVE SEGMENTATION AND VQA RESULTS

The following tables report the quantitative values used in Figure 4 in the main text. Each row represents the mean and standard deviation of multiple runs that differ only for the random seed.

Table 2: **Unsupervised segmentation.** Object attention maps are first upsampled to the input resolution and then binarized individually using Otsu thresholding, which means multiple masks can include the same pixel. Each ground-truth object mask, except the background, is matched with a binarized object attention map. IoU values are first averaged between all objects of an image, then averaged across all images. Results are reported as the mean and standard deviation of several runs. To be compared with Figure 4 (left) in the main text.

| Group | Loss | Mean | Std | Count |
|---|---|---|---|---|
| Cross | Contrastive All | 13.9 | 4.3 | 4 |
| | Contrastive Image | 18.3 | 4.1 | 4 |
| | Cosine Similarity | 19.7 | 1.6 | 4 |
| Slot | Contrastive All | 30.1 | 10.3 | 8 |
| | Contrastive Image | 38.1 | 7.2 | 8 |
| | Cosine Similarity | 43.0 | 3.1 | 6 |
| Slot (Objects only) | Contrastive All | 6.0 | 1.7 | 6 |
| | Contrastive Image | 8.0 | 4.1 | 12 |
| | Cosine Similarity | 5.8 | 3.5 | 4 |

Table 3: **VQA Linear Probes.** A linear probe attached to the output(s) of the model is trained to predict 96 binary questions about the objects present in the image using 10000 samples that were held out during self-supervised training. We report Average Precision on a different set of 5000 samples which were excluded from self-supervised training and linear probe training. Average Precision is computed over the entire set using micro averaging, i.e. flattening all $96 \times 5000$ (target, prediction) pairs before computing precision/recall thresholds. Class imbalance is accounted for by reweighting positive labels. Results are reported as the mean and standard deviation of several runs. To be compared with Figure 4 (right) in the main text.

| Group | Loss | Mean | Std | Count |
|---|---|---|---|---|
| Global Only | / | 71.9 | 9.2 | 43 |
| Cross | Contrastive All | 80.7 | 2.5 | 4 |
| | Contrastive Image | 80.6 | 1.4 | 4 |
| | Cosine Similarity | 80.8 | 2.2 | 14 |
| Slot | Contrastive All | 82.2 | 4.1 | 18 |
| | Contrastive Image | 86.2 | 2.6 | 14 |
| | Cosine Similarity | 85.6 | 2.3 | 20 |
| Slot (Objects only) | Contrastive All | 75.1 | 2.3 | 14 |
| | Contrastive Image | 54.8 | 2.9 | 28 |
| | Cosine Similarity | 51.8 | 1.7 | 14 |

## B.2 QUALITATIVE EXAMPLES OF UNSUPERVISED SEGMENTATION

We include the complete attention maps of the two models portrayed in Figure 3. In these figures, the attentions for all 11 object slots are depicted, along with binarized masks. The columns are arranged so that each attention map is aligned with the closest ground-truth mask. At the bottom of each colum are reported per-slot IoU values.

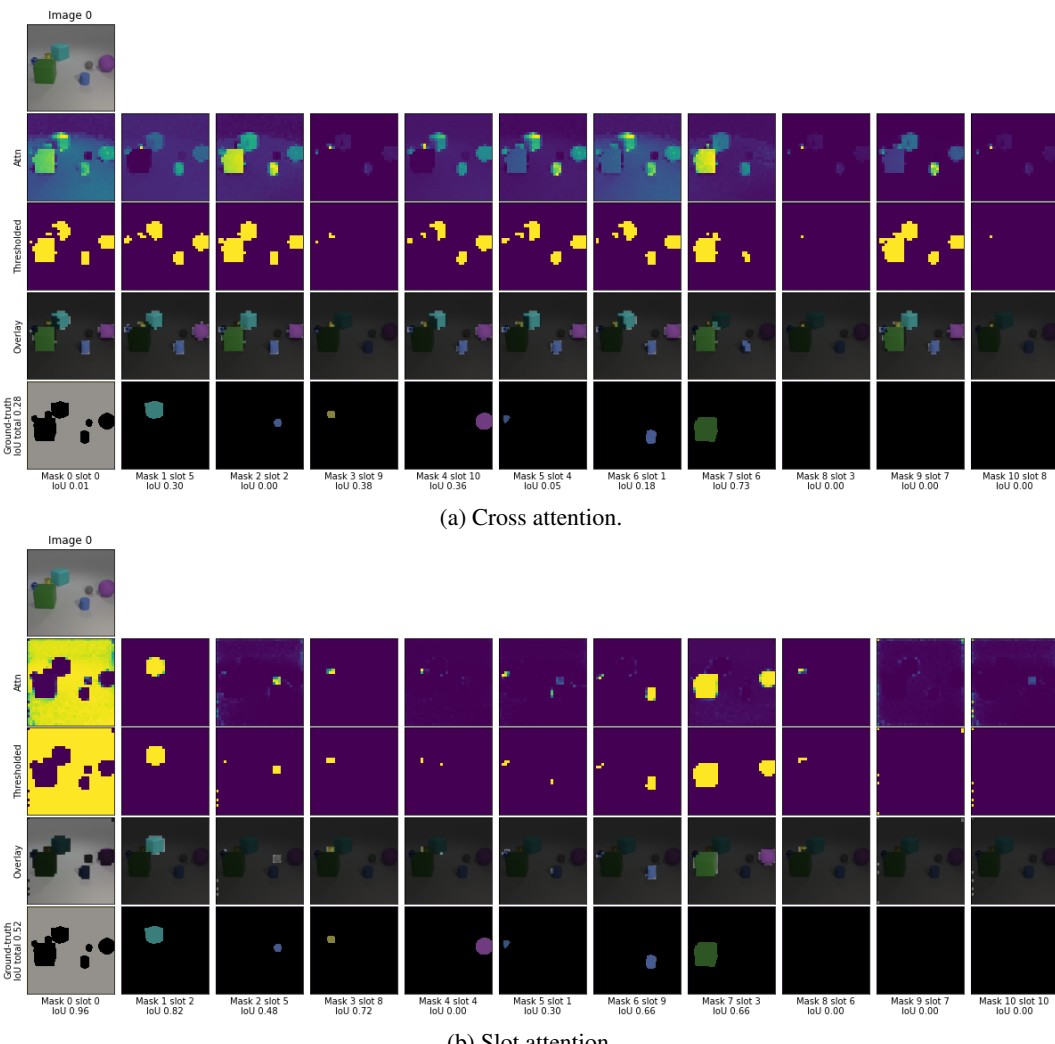

(a) Cross attention.

(b) Slot attention.

Figure 5: First row: per-object attention maps obtained from the first layer of Cross Attention or the first iteration of Slot Attention. Second row: attention masks binarized with Otsu thresholding. Third row: overlay of binarized masks and input image. Last row: ground-truth masks of the 7 visible objects and Intersection over Union (IoU) with the corresponding predicted segmentation. Both methods clearly focus on objects. However, cross attention does not encourage specialization therefore multiple object queries attend to the large cubes in the foreground. Slot attention produces a cleaner segmentation, but tends to merge together spatially-separated objects, e.g. mask 7. A selection of these slots-based segmentation masks is shown in Figure 3 in the main text.

## B.3 CROP SCALE

The following tables provide evidence for the observations in the last paragraph of Section 3. In particular, no degradation is observed when using a wide range of crop scales, e.g. from 10% to 100% of the original area. On the contrary, runs for which the crop scale is very limited, e.g. 10-30%, achieve the lowest scores, suggesting that for this dataset it is more important to sample a diverse set of scene crops for global contrastive learning than to minimize the possibility of false positive in the matching object loss.

Table 4: Unsupervised segmentation and linear-probe VQA performances on a grid of (min, max) values for random crop augmentation.

Count of runs with a given (min, max) crop scale parameters.

| min/max | 30 | 50 | 80 | 90 | 100 |
|---|---|---|---|---|---|
| 10 | 32 | 28 | 14 | 18 | 44 |
| 30 | | 14 | 16 | 12 | 94 |
| 50 | | | 12 | 12 | 52 |
| 80 | | | | 6 | 49 |
| 90 | | | | | 22 |

Mean Intersection over Union of runs with a given (min, max) crop scale parameters.

| min/max | 30 | 50 | 80 | 90 | 100 |
|---|---|---|---|---|---|
| 10 | 13.7 | 29.1 | 34.0 | 35.5 | 35.9 |
| 30 | | 26.3 | 32.9 | 34.8 | 34.0 |
| 50 | | | 30.3 | 34.6 | 33.2 |
| 80 | | | | 25.1 | 19.0 |
| 90 | | | | | 15.4 |

Mean Average Precision of runs with a given (min, max) crop scale parameters.

| min/max | 30 | 50 | 80 | 90 | 100 |
|---|---|---|---|---|---|
| 10 | 75.9 | 84.0 | 86.0 | 85.5 | 84.5 |
| 30 | | 82.9 | 83.7 | 83.8 | 82.2 |
| 50 | | | 80.6 | 81.5 | 78.7 |
| 80 | | | | 72.1 | 66.1 |
| 90 | | | | | 65.3 |

## C   RELATED WORK

This appendix complements the related works discussed in the main text.

### C.1   SELF-SUPERVISED LEARNING.

**Pretext tasks.**   Label-free tasks are used as a pretext for training unsupervised feature extractors on large-scale datasets, e.g. natural images from the internet. These models can then be fine-tuned for specific tasks where data is more scarce. A few examples of pretext tasks include: predicting relative position of image patchs (Doersch et al., 2015), image colorization (Zhang et al., 2016), solving spatial jigsaw puzzles (Noroozi & Favaro, 2016), contextual inpainting (Pathak et al., 2016), predicting orientations (Gidaris et al., 2018).

**Latent-space losses.**   Pretext tasks such as reconstruction, colorization and inpainting, rely on reconstruction losses in pixel space, which may be challenging to optimize. A popular alternative is to define losses directly in the latent space, giving rise to contrastive, similarity, and self-distillation approaches. These approaches can be categorized as requiring a momentum-updated teacher (He et al., 2020; Grill et al., 2020) or performing online self-distillation (Chen et al., 2020; Chen & He, 2021; Caron et al., 2021). Based on the training objective, some methods use explicit negative samples (He et al., 2020; Chen et al., 2020), or only positive samples (Grill et al., 2020; Chen & He, 2021), or even a negative-free clustering-based task (Caron et al., 2020; 2021).

### C.2   OBJECT DISCOVERY.

Kipf et al. (2020) learns structured world models Contrastive learning structured world models using CNN features and applies a triplet loss between corresponding slots in subsequent time frames, without requiring bipartite matching. Compared to our approach, the architecture does not contain an explicit object extraction layer and no mechanism encourages diversity across objects.

Racah & Chandar (2020) also pools object representations from CNN features, which requires an additional slot diversity loss to encourage representing distinct objects. The learning signal is provided by a temporal contrastive loss across video frames.

Löwe et al. (2020), similar to the above, focus on discovering objects in videos with a next-frame prediction contrastive approach. However, the loss is defined on a global frame representation instead of using tokens extracted with slot attention.

