# OpenReview forum: "Towards Self-Supervised Learning of Global and Object-Centric Representations"
_ICLR.cc/2022/Workshop/OSC — ICLR2022 OSC  Poster_

### Official Review · Reviewer_MmxT · 2022-03-06
**An interesting paper combining Slot Attention and contrastive learning**

**Rating:** 2
**Confidence:** 3

**Review:**

This paper builds upon two recently popular fields of work, the object-centric representation learning and the self-supervised contrastive learning. Instead of using the pixel-wise reconstruction loss as the main supervision signal, this paper introduces both image-level and object-level contrastive loss to help object discovery. The segmentation masks can be extracted as the attention masks in Slot/Cross Attention. The trained model achieves reasonable performance on segmentation IoU and VQA AP. More comments are as follows:

- I completely agree with the author that, per-pixel reconstruction loss might hinder the model to scale up to real-world images. I am very happy to see the contrastive loss from SimSiam can work here. Simply maximizing the similarity of positive pair is indeed an easy-to-implement and promising technique
- I am a bit surprised by the fact that the global loss matters that much, and I am interested in how the segmentation masks look like in this case. After all the object loss is still minimized, especially in the CTRALL case
- The IoU result (40%) is not very good. The Slot Attention paper only has numbers for ARI. But on the other hand, object-centric representation learning doesn't necessarily require segmentation. The learned good object representation is verified by the VQA task
- It would be very interesting to extend this work by incorporating with techniques in unsupervised video representation learning papers

Overall, I believe this paper presents an interesting direction of object-centric representation learning. I vote for the acceptance of this paper.

---

### Official Review · Reviewer_VY15 · 2022-03-15
**comments for TOWARDS SELF-SUPERVISED LEARNING OF GLOBAL AND OBJECT-CENTRIC REPRESENTATIONS**

**Rating:** 2
**Confidence:** 3

**Review:**

This paper proposes a method to learn object representation and train the model in a self-supervised manner with a contrastive loss. Ablation studies involving different kinds of attention maps and object losses are conducted to prove the efficacy. However, why isn't there any baselines in the experiment section to compare with? It is necessary to show whether your method outperforms other state-of-the-art models, for example, the vanilla self-supervised learning methods of contrastive learning and simCLR.

---

### Decision · Program_Chairs · 2022-03-23

**Decision:**

Accept (Poster)

**Comment:**

The reviewers agree the paper should be accepted at the workshop. Congratulations!

It would be great to include baselines, as pointed out by reviewer VY15. If this is not feasible for this workshop's camera-ready version due to the time limitations, we highly recommend including them in future iterations of this work.